# A Streamlined Approach to Pathway Analysis from RNA-Sequencing Data

**DOI:** 10.3390/mps4010021

**Published:** 2021-03-17

**Authors:** Austin Bow

**Affiliations:** Department of Large Animal Clinical Sciences, University of Tennessee, Knoxville, TN 37996, USA; abow@vols.utk.edu; Tel.: +1-865-963-1952

**Keywords:** transcriptomics, data processing, RNA-sequencing, network, mapping, cytoscape, enrichment analysis, database, protocol

## Abstract

The reduction in costs associated with performing RNA-sequencing has driven an increase in the application of this analytical technique; however, restrictive factors associated with this tool have now shifted from budgetary constraints to time required for data processing. The sheer scale of the raw data produced can present a formidable challenge for researchers aiming to glean vital information about samples. Though many of the companies that perform RNA-sequencing provide a basic report for the submitted samples, this may not adequately capture particular pathways of interest for sample comparisons. To further assess these data, it can therefore be necessary to utilize various enrichment and mapping software platforms to highlight specific relations. With the wide array of these software platforms available, this can also present a daunting task. The methodology described herein aims to enable researchers new to handling RNA-sequencing data with a streamlined approach to pathway analysis. Additionally, the implemented software platforms are readily available and free to utilize, making this approach viable, even for restrictive budgets. The resulting tables and nodal networks will provide valuable insight into samples and can be used to generate high-quality graphics for publications and presentations.

## 1. Introduction

With the increased availability and relatively low cost of modern RNA-sequencing (RNA-seq), the restricting factors in utilizing such an assessment tool have shifted from financial budgeting to data processing time. This is primarily due to the sheer quantity of data generated by the comparison of even a single experimental sample against its respective control, which is only further compounded when examining multiple experimental sample groups. The resulting challenge therefore becomes how to sift through the data for comparative elements that contain relevant and meaningful information about the submitted samples. Fortunately, some organizations that offer RNA-seq services also provide a rudimentary analysis of the output data based on certain databases. For example, the company Novogene includes in its services a comprehensive report that identifies pathways of interest (PoIs) based on pathway enrichment using KEGG database pathways and potential gene ontology (GO) relations. However, this report may not capture all PoIs, particularly if the utilized databases are limited, and thus, a more extensive examination of the data may be necessary. For this reason, such reports serve ideally as a foundation for further evaluation of the normalized differentially expressed gene values. This will not only allow for the fine tuning of relevant pathways and systems, but will also permit the generation of graphics and tables that best illustrate important comparative characteristics. This cross-referencing of normalized data using multiple enrichment tools serves to bolster both the accuracy and extent of detail for the resulting conclusions. While a wide array of such software tools exists for enriching gene sets against given databases, the more user-friendly options will be the focus of this article. Furthermore, only those tools that are readily available and free to users will be examined, since paid software can substantially inflate budget requirements for a study and may not necessarily provide the desired output, i.e., specific relationships or polished graphics. The data processing flow described in this work will be based on data generated by Novogene; however, the methods should be applicable for any large gene expression data set. It should be noted that the described protocol operates on normalized datasets for differentially expressed genes, and therefore, may not be applicable for research focused on gene discovery in nonmodel organisms or phylo-transcriptomics, which require examination of raw RNA-seq data [1,2]. The most vital aspect of this processing plan will be to clearly define the extent of pathway searching within the context of a specific set of research questions and confine information mining to these parameters. This is essential, as a common issue in assessment of massive data pools is the development of conclusions that inaccurately portray the relationship between samples due to selection bias and the cherry-picking of obscure pathways that strengthen preconceived conclusions. For example, if a study intends to assess an experimental material that is anticipated to enhanced cellular adherence, the scope of at least the initial data processing should be limited to pathways related to this biological function, and extraneous connections should be logged as potential PoIs for later examination. This can prevent developing correlations that obscure the original hypothesis.

## 2. Experimental Design

### 2.1. Data Processing

Data output from transcriptomic-based analyses like RNA-seq can initially appear intimidating due to file size and complexity. Therefore, the formation of a central set of research questions to establish general parameters for pathway and gene ontology (GO) selection is a critical initial step. This collection of hypotheses, hereafter denoted as the focal parameter set, will encompass the expectations of relations that will be examined between sample groups to narrow the focus of downstream pathway enrichment and mapping functions. Following initial data assessment with a single focal parameter set, this process can be repeated with new foci to examine intergroup relations associated with different biological processes. Segmenting the evaluation of the original data in this manner can help to narrow the focus of individual analyses and minimize off-target selections, which may result in fishing expeditions (also known as multiple testing) [3]. For example, the exploration of complex disease pathways may not be relevant for data produced by cell monolayer samples, so selecting these pathways during downstream analysis could lead to misleading interpretations.

Apart from focal parameter set selection, the formatting of data also represents an essential pre-analysis step, as this divides initial information into more manageably sized files and helps to fine-tune comparative assessments downstream. A simple method for this is the separation to three discrete datasets based on gene fold expression values, dividing the initial normalized gene data to categories of upregulated, static, and downregulated expression. The upper and lower threshold of the “static” subset will, in some cases, be predetermined if the data is being provided by a commercially available processing source, but will be largely dependent on the extent to which the investigator wants to scrutinize data bands [4,5]. For the purpose of displayed examples, this threshold has been set to an increase or decrease of two-fold difference for the static expression dataset, based on Novogene’s suggestion. Gene relations above and below these thresholds are considered to be up- and down- regulated, respectively. The isolation of these independent groups to corresponding data spreadsheets will allow for convenient and rapid access. As a final formatting step, attention should be given to the identifier type available for the gene sets. As Ensembl and Entrez IDs are the most common input methods for gene data entry on pathway enrichment and mapping software platforms, converting the current gene identifiers to one or both of these formats will help to improve software recognition. Conveniently, there are multiple readily available conversion tools for gene IDs, with both the software DAVID (Database for Annotation, Visualization, and Integrated Discovery) and SynGo online platforms providing a means to generate complimentary lists of Ensembl and Entrez IDs for input gene lists, thereby providing a rapid means for establishing these common identifiers for data [6,7,8].

Though there are a multitude of enrichment software packages available, with some allowing free access and others requiring purchase, many operate based on similar collections of pathway and GO databases. It is therefore important to become familiar with some of the more commonly utilized of these databases in order to understand the limitations of downstream analyses, as well as to be able to determine the optimal software for assessing a given dataset and focal parameter set. The following described databases represent some of the more robust and commonly referenced sources. Critically, some of these are limited in information that would be required to adequately characterize certain species or systems, resulting in insufficient pathway/gene ontology data. A basic understanding of these core databases can therefore help to minimize loss of time and funds during the processing of datasets.

The Kyoto Encyclopedia of Genes and Genomes (KEGG) is a massive pathway database that includes compounds, reactions, and full pathway maps for a wide array of species [9,10]. This particular database is utilized by many enrichment programs and should be considered an essential reference for any pathway analysis. Other pathway databases, such as WikiPathway, Pathway Integration Database (PID), and Reactome, can also serve as useful reference material, particularly when examining some disease models [11,12,13,14]. Apart from defined pathway references, examination of gene ontology (GO) is similarly critical for these types of studies and works to correlate gene sets to biological processes [15,16]. Knowledge of these databases and their limitations can greatly improve the ability to select an appropriate enrichment software, which, in turn, enhances the accuracy of downstream data exploration.

### 2.2. Pathway Analysis Tools

The selection of enrichment software platforms best suited for the established gene set lists and their defined focal parameter set can be challenging, as differing reference databases can result in variations for generated pathways of interest (PoI) lists. For this reason, it is highly beneficial to implement multiple enrichment tools for each individual dataset and cross-reference these output data to elucidate common PoIs, which will serve as primary targets for analysis. These target PoIs can then be further examined and mapped to develop conclusions and graphics. The following are some of the common and user-friendly software tools that can be utilized for effective pathway analysis. In addition to these described tools, there are a wide array of other user-friendly tools such as FunRich and GSEA-MSigDB that are not further covered in this protocol [17,18]. These pathway analysis platforms can be explored and implemented alongside or in-place of those described here to optimize this workflow for the target experiment. Apart from the ConsesusPathDB (CPDB) program, the output of these tools will be largely processed in data spreadsheets, with elements requiring sorting based on significance and relevance.

The Integrated Molecular Pathway Level Analysis (IMPaLA) program offers a user-friendly and rapid means for generating pathway lists associated with a specific gene list, either through entry of genes IDs alone or with associated expression fold values [19,20,21]. Additionally, this software provides the potential to overlap these pathways with related pathways from input metabolite data, though this is beyond the scope of this work and will not be further discussed. The input gene IDs can be in several forms, including both Ensembl and Entrez formats, and can be directly pasted as text from spreadsheets along with corresponding values. The produced pathway list from running the Wilcoxon pathway enrichment analysis can be then downloaded as a data spreadsheet and edited to select PoIs.

The KEGG Orthology-based Annotation System (KOBAS) program offers a means for rapidly generating pathway list tables from gene enrichment data, accepting Fasta Nucleotide Sequences, Ensembl ID, Entrez ID, Gene Symbol, and NONCODE ID formats [22]. The software operates based on annotation and pathway network information from the KEGG database, thereby offering the ability to assess data from a large number of species, whereas many databases are restricted to pathways and GO for human, mouse, and yeast systems [23,24,25]. In addition to gene enrichment functions observed in other software platforms, the same gene list can be further modified through the KOBAS *Annotate* application. This can provide beneficial insight into the related pathways and Gene Ontology connections associated with the set. The data output for gene enrichment includes pathways with corrected *p*-values and the list of associated genes. These data are presented in a text file that can be downloaded and imported into a spreadsheet for editing.

As with the previously discussed software, the Database for Annotation, Visualization, and Integrated Discovery (DAVID) program offers a means for generating pathway and GO tables from gene lists. The data output is robust compared to similar enrichment software platforms, with DAVID offering the ability to enrich data using both KEGG and Reactome databases. Heatmaps are also generated, which can aid in visualization of the genes associated with certain pathways [26,27,28]. The pathway and GO lists can then be downloaded and imported to a spreadsheet. In particular the false discovery rate (FDR) column will be most applicable for establishing lists of significant pathways and Gene Ontology for the three primary gene lists. As FDR values represent the statistical likelihood that a comparison with a statistically significant *p*-value is accurate, these values are essential for selection of PoIs [29].

The Consensus Path Database (CPDB) program offers one of the most robust tools for assessing gene and metabolite data sets, including an effective network graphic generator with a user-friendly input [30,31,32,33]. The gene analysis section of the site includes both over representation and enrichment analysis methods that can be utilized to evaluate PoIs associated with a data set. Gene IDs in Ensembl, Entrez, Gene Symbol, or Uniprot formats can be entered to generate a list of potential target pathways, with corresponding corrected *p*-values and the source database. One of the key features that separates CPDB from other enrichment software platforms is the board spectrum of integrated databases, which are listed on the home page of the platform. This wealth of reference material substantially enhances the accuracy for detecting significantly impacted pathways in experimental data; however, it should be noted that CPDB analysis is limited to human, yeast, and mouse genomes. The initial list of generated pathways, in contrast with previously discussed software platforms, is expressed as an interactive webpage that can be sorted based on significance, with the option to then select relevant pathways by checking associated boxes. The selected PoIs can then be visualized in an editable network graphic that displays the interconnectivity among the chosen pathways. These networks can then be saved as standard image files using the print screen function after organizing nodes and cropping accordingly. The PoIs and their associated significance values can then be cross-referenced with the output from other software tools, with pathway interconnectivity graphics offering supplemental information.

The Cytoscape software platform represents a vast array of built-in tools and databases for constructing and visualizing nodal networks [34,35,36,37,38,39,40,41,42]. Additionally, the program can be outfitted with a wide variety of plug-ins for enriching, fine-tuning, and polishing output networks. The basal software can be downloaded from the Cytoscape website, with plug-in applications available for download from both this site and through the in-program *App Manager* function [43]. These applications, in addition to the core tools available in Cytoscape, allow for rapid transformation of input fasta files into well annotated network maps that can be further polished for use as publication and presentation graphics. To accomplish this, the organism-specific biological data available through the BioGrid database can be utilized to generate basal networks into which experimental data can be imported [44]. This not only overlays input data with a defined nodal network, but also assigns basic annotations for biological functions and pathway relations. These generated networks for individual gene sets can then be manipulated and distilled to highlight relations and functional groupings of interest.

### 2.3. Nodal Network Generation

One of the key complicating factors associated with generating transcriptomic-based networks is scale. Attempts to input all available data can rapidly lead to massive collections of nodes and pathways, obscuring relevant information and making downstream analysis and presentation of results highly difficult. Additionally, these large files will require increased load time and can cause software crashes, all of which further emphasize the importance of separating data to manageable portions. This can best be accomplished by dividing data into the previously discussed pools of upregulated, downregulated, and static gene expression categories, which will allow the assessment of intra-pool gene sets and pathways. Conclusions regarding impacted biological pathways and functions can then be drawn based on the differences observed among these groups [45,46].

An effective technique for developing networks from collections of differentially expressed genes is to utilize a preconstructed network for the target organism onto which experimental data can be integrated. As previously mentioned, datasets available through the online source BioGrid offer invaluable basal pathway configurations for a wide variety of species and can be readily downloaded and imported into Cytoscape. Once the basal network has been established, experimental data can then be integrated with the BioGrid information to produce a filtered network of target nodes that maintain pathway interconnective elements. Repeating these steps for each of the defined gene sets will yield network files that accurately represent the connective pathway links present in the experimental data.

The raw nodal networks generated through the previously described steps, though comprehensive in displayed data, often require a level of modification to develop graphics that are aesthetically appealing and can be utilized in publications. Cytoscape readily allows for the inclusion of both image- and text-based annotations, network color-coding, and node positional arrangement, all of which will enable network clean-up and enhance the readability of output graphics. First and foremost, the necessary magnification and resolution of the downstream graphics must be determined. For figures intending to display general pathway information, magnification of the network can be substantially reduced, whereas graphics meant to indicate interactions of specific genes must be magnified to at least an extent to make gene names legible. When attempting to increase resolution of pathway data through magnification, excess network data may need to be pruned and nodes rearranged to best display the interactions of interest without obstruction. In cases where additional annotation is not required, these graphics can be captured using the export function in Cytoscape, which will generate a PNG file (default) of the current section of the on-screen network (it is important to note that only the information on-screen for the network will be exported as an image).

To further enhance the appearance of networks, the stringApp application, which is a plug-in for Cytoscape, can be used to integrate the existing network with protein-protein interaction data [47,48]. This application also converts the standard Cytoscape node design to a graphically appealing protein ribbon structure overlaid on marble-like image. Additional plug-ins such as CyAnimator, offer the ability to define frame-by-frame pans through a network that can be exported as a high-resolution video file [49]. This is particularly effective for presentation graphics that are intended to step an audience though a general process.

Apart from these specialized applications, the use of background graphics and other annotation tools for network maps can be a highly effective method for data organization. Implementation of blank plot formats onto which network data can be overlain offer a means for expressing output data with more context than possible with an unannotated network. This can be further enhanced by the adding color gradients, located in the *style* panel of the software, which increases visibility of impacted pathways and relations.

## 3. Procedure

### 3.1. Data Processing

Locate and open the files containing the differentially expressed gene lists for a sample comparison.Copy and paste provided gene identifiers directly into SynGO conversion tool [7] and select corresponding identifier type and species from drop-down menus.Press *Start ID Conversion* and download resulting table to text file.Import text file to new spreadsheet and integrate data with differentially expressed gene list document resulting in enhanced gene identifier options for downstream enrichment programs.Establish expression thresholds for gene sets to clearly define upregulated, downregulated, and static expression sets (this can be based on pre-existing thresholds provided by company or modified based on relevant literature).Segment master gene list to three separate spreadsheets based the classification of whether the gene is upregulated, downregulated, or statically expressed between the experimental group and respective control in comparison.Repeat steps 1–6 for all comparisons to be examined.For each comparison set, determine biological functions/pathways most likely to be impacted based on the overarching hypothesis for the study (i.e., samples derived from cells treated with a compound intending to enhance osteogenesis would be examined focally for variations in osteoblastic differentiation or cellular adhesion pathways) and classify these functions/pathways as the focal parameter set.Record the established focal parameter set for use in later enrichment steps.Open a blank spreadsheet and designate as “Enrichment Analysis Master List” (this will be the repository for downstream enrichment data results from pathway analysis tools).

### 3.2. Pathway Analysis

#### 3.2.1. IMPaLA Analysis

Open IMPaLA homepage [50].Input gene identifiers and respective expression values, select gene identifier type from drop-down menu, select *Wilcoxon pathway enrichment analysis*, and press *Start Analysis*.Verify that an adequate percentage of the genes were detected and mapped (a low percentage may indicate input error).Sort resulting table by *Qgenes* and download pathway list as a spreadsheet; see Figure 1.Remove list items above designated significance threshold based on *Qgenes* value and transfer remaining pathways to “Enrichment Analysis Master List”.

#### 3.2.2. KOBAS Analysis

Open KOBAS homepage [51] and navigate to gene enrichment analysis (*Enrichment* → *Gene-list Enrichment*).Enter target species, select gene identifier type for menu (note that KOBAS is capable of accepting Fasta type files), input gene identifiers, select *KEGG Pathway* (*K*), *Reactome* (*R*), *GO* (*G*) database options, and press *Run*.Sort table items by corrected *p*-value and download list using *Download Total Terms* function to generate a text file that can be imported to “Enrichment Analysis Master List”; see Figure 2.Click *Visualization of Filtered Terms* (note that this function is currently in demo state and will likely be improved in the future) to generate circular network, barplot, and bubble plot representations of data.Save graphics of interest with the save feature at bottom right of each image (these graphics can be stored for supplemental data).

#### 3.2.3. DAVID Analysis

Open DAVID homepage [52] and navigate to gene enrichment analysis (*Start Analysis* → *Upload*).Input gene identifiers, select identifier type from drop-down menu, specify list as *gene list*, and press *Submit List*.Select species to filter output data (In *Gene List Manager* tab select the target species name → click *Select Species* → for *Select List to*: click *Use*)Select *Functional Annotation Tool* and utilize DAVID default search criteria (Search criteria can be modified to further expand pathways/biological functions examined. To edit the search parameters, such as utilized pathway databases, expand annotation categories and select/deselect search parameters).Open *Functional Annotation Clustering* and download cluster data to text file for importing to “Enrichment Analysis Master List”; see Figure 3.For saving heatmaps of high enrichment score clusters, open heatmap of interest and print resulting webpage to PDF for supplemental data file.

#### 3.2.4. CPDB Analysis

Open CPDB homepage [53] and navigate to gene enrichment analysis (*Gene Set Analysis* → *Enrichment Analysis*).Input gene identifiers and respective expression values, select identifier type from drop-down menu, and press *proceed*.Verify that an adequate percentage of the genes were detected and mapped (a low percentage may indicate input error).Select *1-next neighbor, pathways as defined by pathway databases, gene ontology level 2 categories*, and *sets of genes whose products are found together in protein complexes* from the following page to establish the extent of the enrichment analysis.Sort the resulting list by *q-value* and select pathways/biological functions relating to focal parameter set; see Figure 4.Download selected pathway/biological functions to text file and import to “Enrichment Analysis Master List”.On CPDB, press *visualize selected sets* to generate a nodal network detailing the connective elements among selected pathways/biological functions; see Figure 5.Modify node position and connective edge filter settings to best display data.Select *graph legend* to insert legend onto network image.Capture network image with screen print function and edit/save final graphic using PowerPoint software.

#### 3.2.5. Enrichment Analysis Cross-Referencing

Organize data within “Enrichment Analysis Master List” spreadsheet so that pathways/biological functions and respective corrected significance values from IMPaLA, KOBAS, DAVID, and CPDB are in side-by-side lists.Sort each list by corrected significance value and remove all line items above designated significance threshold.Based on focal parameter set, highlight all PoIs using the cell highlighting tool in spreadsheet application.Sort each list by cell color (this will bring all highlighted cells to the top of lists) and remove unhighlighted line items.Manually compare lists for commonalities and highlight with a new color any PoIs shared by two or more pathway analysis tools (using a different color to denote that a PoI is common to two, three, or four pathway analysis tools).Generate a new document designated as “Ranked PoIs” categorizing PoIs by commonality among pathway analysis tools and sorted by corrected significance values.Repeat steps 3.2.1–3.2.5 for the upregulated, downregulated, and statically expressed gene lists for each comparison being assessed.For each set of upregulated, downregulated, and statically expressed gene lists, construct composition spreadsheet for top-ranked PoIs (those observed across multiple pathway analysis tools) that includes a column of gene symbol names for all genes associated with selected PoIs and columns headed with each PoI name for developing a search index in Cytoscape (entering YES in the PoI name column for genes associated with it will permit rapid categorization downstream).

**Figure 4 mps-04-00021-f004:**
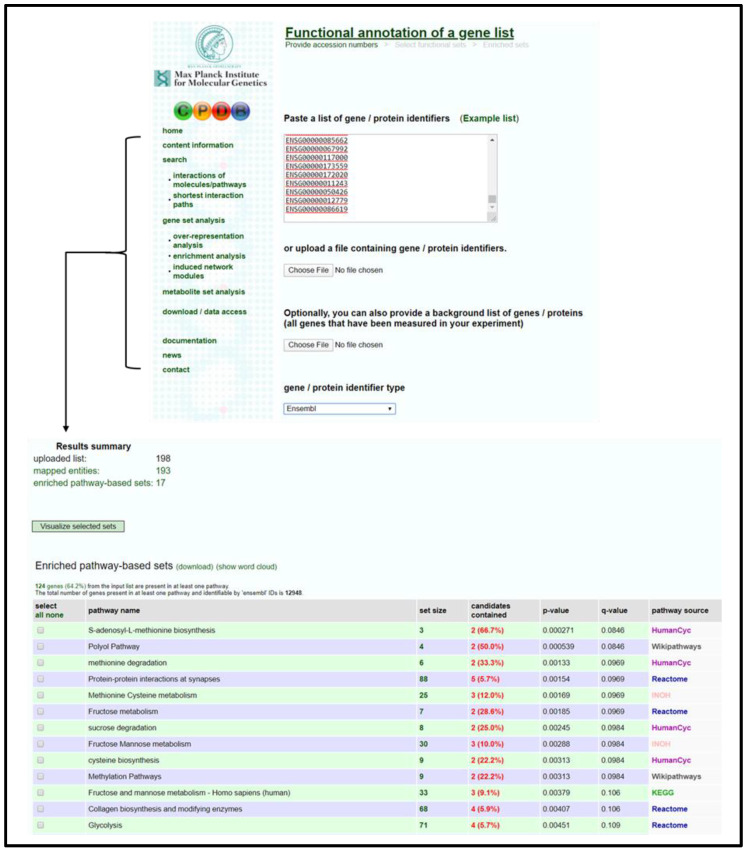
CPDB gene set analysis over-representation page and resulting output data table for gene list. Data provided in the output table (from left to right) are pathway name, total number of genes in pathway, the number of input genes in pathway, *p*-value associated with number of input genes involved in pathway, corrected significance value that accounts for false discovery rate, and the source database for the pathway. The left-most column of check boxes can be selected to determine which pathways will be visualized in the generated network graphic.

**Figure 5 mps-04-00021-f005:**
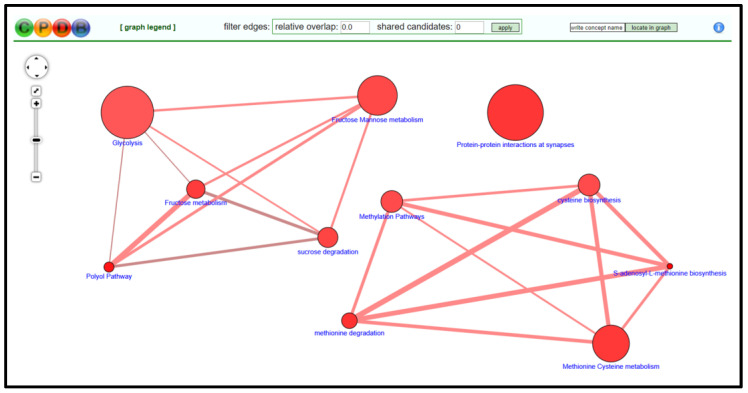
Example of CPDB generated network graphic based on manually selected pathways of interest. Nodes can be organized to most effectively display pathway interconnectivity data. Furthermore, “filter edges” options allow for restricting pathway connections.

#### 3.2.6. Cytoscape Mapping

Open Cytoscape homepage [54] and download the most recent version of Cytoscape software (can periodically check homepage for updated versions if already downloaded).Install stringApp application through in-software or online app manager (other plug-ins can be install and explored for effectiveness in modifying data).Open BioGRID homepage [55] and download the most recent zip file for “BIOGRID-ORGANISM-(Update Number).psi25”.Import BioGRID Homo Sapiens XML document from zip file into Cytoscape (*File* → *Import* → *Network from File* → Select Homo Sapiens XML Document); see Figure 6.Import upregulated, downregulated, and statically expressed gene lists for target comparison and overlay with BioGRID basal network (*File* → *Import* → *Table from File* → Select Gene List Name → Select *To selected networks only* for *Where to Import Table Data* → Select BioGRID basal network from *Network List* → Select *Node Table Columns* for *Import Data as* → Select *Short Label* for *Key Column for Networks* → Keep *Case Sensitive Key Values* selected → Find the gene symbol name column of gene list and click on column to edit → Set as *Key* with the key shaped icon under *Meaning* in the drop-down menu → Press *OK* to import and overlay data).Import composition spreadsheet for data set and overlay with network for PoI search index (*File* → *Import* → *Table from File* → Select Composition Spreadsheet Name → Select *To selected networks only* for *Where to Import Table Data* → Select BioGRID basal network from *Network List* → Select *Node Table Columns* for *Import Data as* → Select *Short Label* for *Key Column for Networks* → Keep *Case Sensitive Key Values* selected → Find the gene symbol name column of composition spreadsheet and click on column to edit → Set as *Key* with the key shaped icon under *Meaning* in the drop-down menu → Press *OK* to import and overlay data).Highlight gene list data nodes from newly constructed network by selecting them from Node Table (sorting the table by a column of values not in the original BioGRID data such as the entrez ID or HGNC ID will bring all gene list data to the top of table).Generate new network from selected nodes and edges for data-specific nodal network (*File* → *New Network* → *From Selected Nodes, Selected Edges*).Remove duplicate edges (*Edit* → *Remove Duplicate Edges…* → Select target network and press *OK*).Remove self-loops (*Edit* → *Remove Self-loops…* → Select target network and press *OK*).Clone data-specific network to maintain unmodified version of network for reference (*File* → *New Network* → *Clone Current Network*).Add color gradient to editable data-specific network nodes based on associated expression value to establish (*Style* tab → *Fill Color* drop-down menu → Select expression fold change value column for *Column* → Select *Continuous Mapping* for *Mapping Type* → Select colors for low, high, and static expression to form gradient → Press *OK* to apply).Separate color-coded network nodes by associated PoI using search index columns in node table (sort each PoI column to isolate nodes associated with it, then click and drag these clusters to independent regions on the mapping panel).Organize PoI clusters either manually or using one of the automated formatting options under the *Layout* tab.Annotate resulting node map with figures and text as needed using the *Annotation* tab.For protein-protein annotation data and enhanced node graphics, clone the current network (covered in a previous step) and apply the stringApp plug-in to the resulting map (this will replace nodes with high resolution graphics of the associated protein structure and add protein-protein data to network annotation, but may reorganize network orientation).Once annotation and organization of nodal network is complete, export network map as image for use in presentations/publications (*File* → *Export* → *Network to Image* → Save as .PNG file for high resolution).

## 4. Expected Results

The described protocol, an overview of which can be observed in Figure 7, will primarily yield a set of three “Ranked PoIs” documents for each comparison being examined, distilling original upregulated, downregulated, and statically expressed gene lists to condensed lists of significantly impacted pathways/biological functions; see Figure 8. These “Ranked PoIs” documents will be further supplemented by heatmap and plot graphics generated by DAVID and CPDB platforms, which can be utilized in presentation and publications to support and convey complex data. Additionally, the described network generation and polishing procedural steps listed for Cytoscape can be employed to create network graphics such as those displaying in Figure 9 and Figure 10. These networks will serve as effective visual representations of pathways/biological functions impacted within a particular comparison group. These example images depict differing magnification and annotation styles, illustrating the various options available for presenting pathway relations.

## Figures and Tables

**Figure 1 mps-04-00021-f001:**
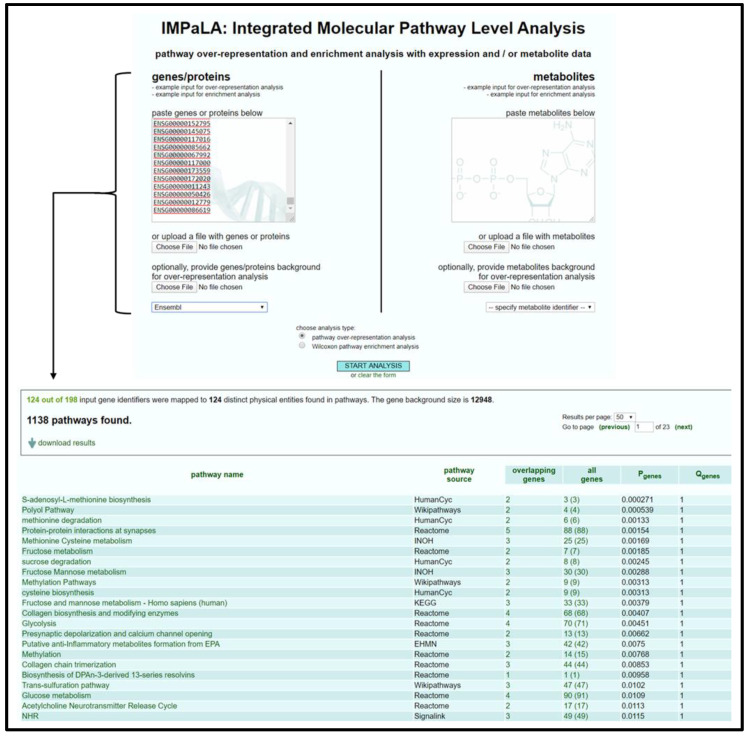
IMPaLA home page and resulting output data table for gene list. Data provided in the output table (from left to right) are pathway name, the source database for the pathway, the number of input genes in pathway, IDs for input genes involved in pathway, total number of genes in pathway, *p*-value associated with number of input genes involved in pathway, and corrected significance value that accounts for false discovery rate.

**Figure 2 mps-04-00021-f002:**
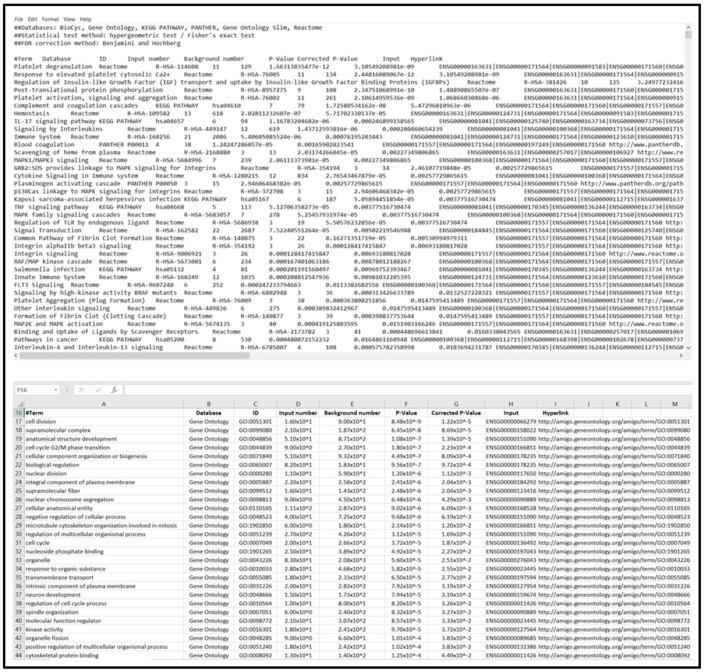
KOBAS data output. Original text file ouput from KOBAS and imported text into spreadsheet for editable file. Provided information (from left to right) are name of pathway/GO, source database for pathway/GO, ID for pathway/GO, number of input genes involved in pathway/GO, number of total genes in pathway/GO, *p*-value associated with pathway/GO, corrected significance value accounting for false discovery rate, IDs of input genes involved in pathway/GO, and hyperlink to source database file for pathway/GO.

**Figure 3 mps-04-00021-f003:**
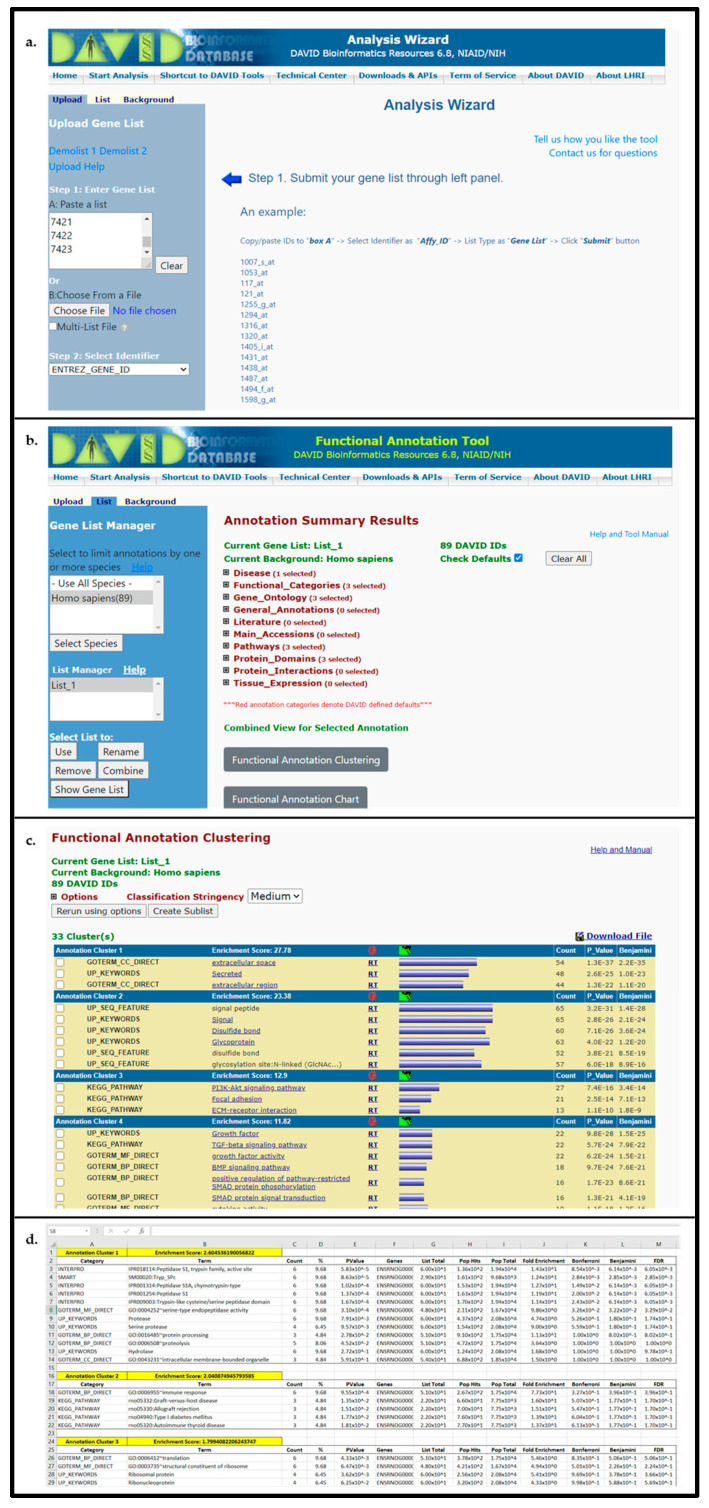
DAVID gene set analysis page (**a**), resulting data overview page (**b**), functional annotation cluster results (**c**), and output text file imported to spreadsheet (**d**). Data provided in the output table indicates enrichment value associated with the annotated cluster, with each cluster detailing a set of biological functions and corresponding significance values.

**Figure 6 mps-04-00021-f006:**
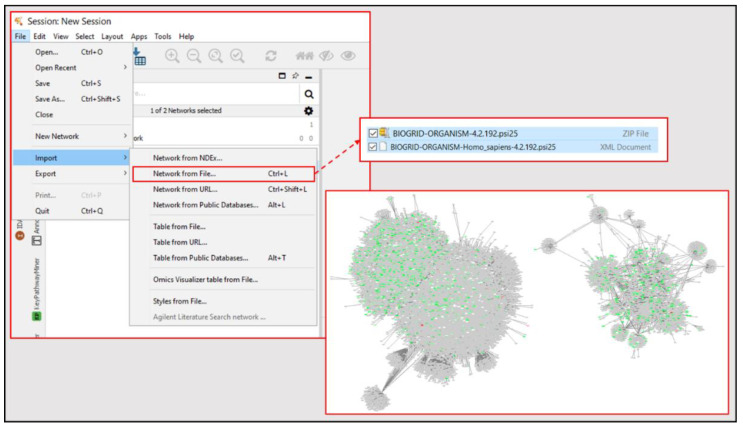
Procedural steps for generating basal network from BioGrid database homo sapiens data set.

**Figure 7 mps-04-00021-f007:**
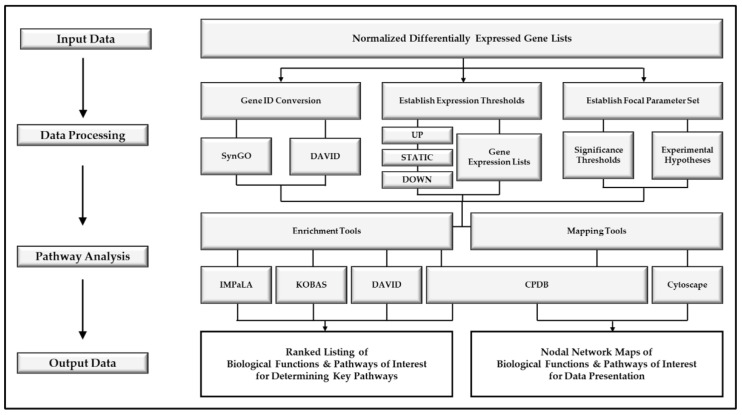
Workflow overview of pathway analysis protocol for RNA-seq data. Input data of normalized differentially expressed genes lists for samples (**Top**) is subjected to *Data Processing* and *Pathway Analysis* steps to generate both ranked lists and nodal network maps of biological functions and pathways of interest (**Bottom**).

**Figure 8 mps-04-00021-f008:**
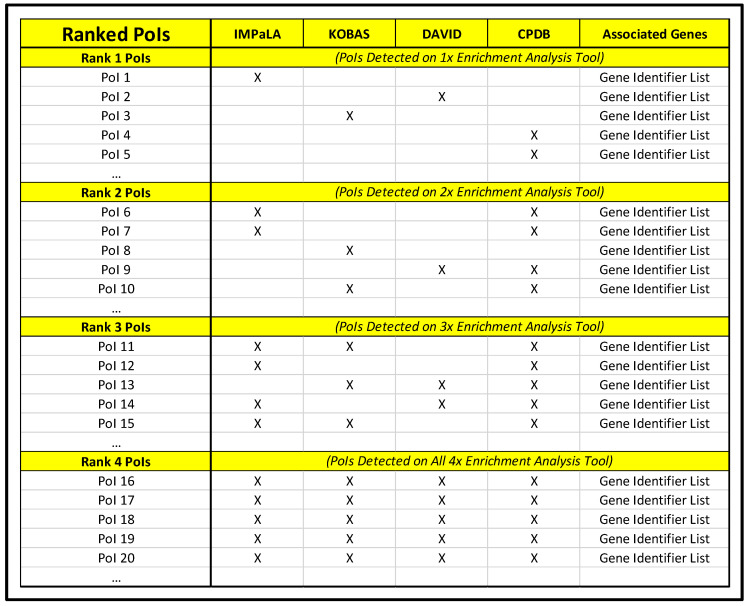
Example of Ranked PoI spreadsheet displaying (From Left to Right) PoI name, enrichment tools used for detection, and a list of genes involved in the pathway/biological function for each rank set.

**Figure 9 mps-04-00021-f009:**
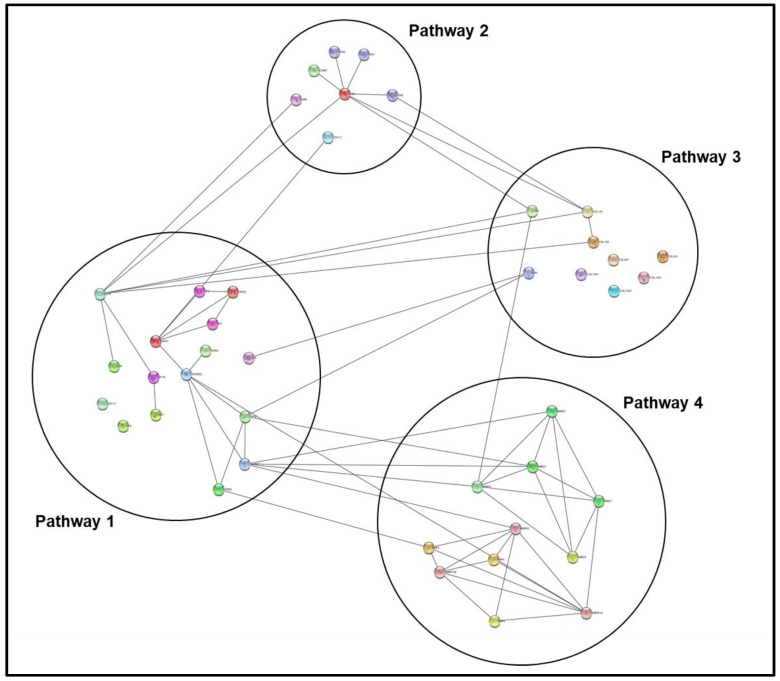
Annotated network with integration of STRING application intended to show general gene numbers detected within associated pathways and interconnective elements between pathways.

**Figure 10 mps-04-00021-f010:**
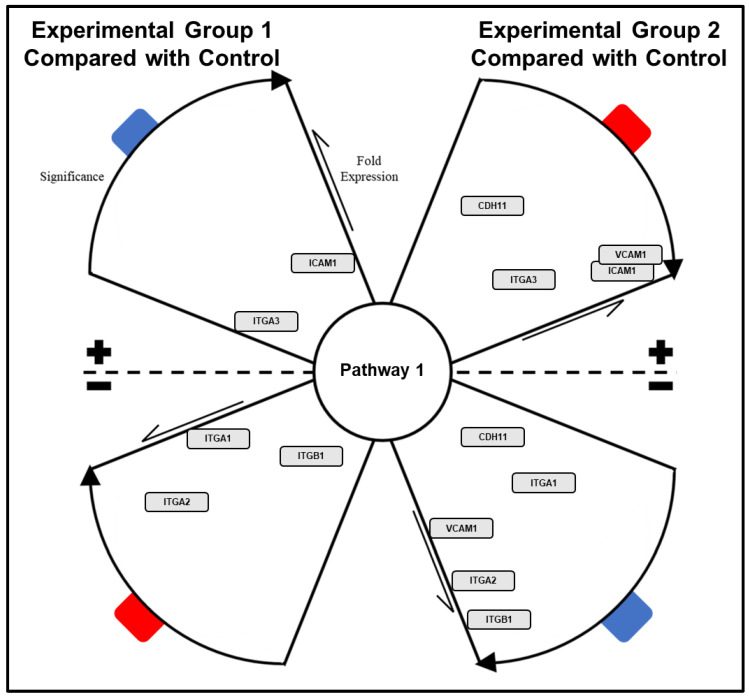
Annotated network data set with pathway associated genes organized within propellor plot diagram for demonstrating expression changes and significance of target genes in multiple experimental groups as compared to a common control.

## Data Availability

Not Applicable.

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
