# Peer review of "A Streamlined Approach to Pathway Analysis from RNA-Sequencing Data"

_mps, 2021, doi:10.3390/mps4010021_

Round 1

Reviewer 1 Report

This is a relatively straightforward manuscript that presents a processing protocol of RNASseq data for gene ontology, pathway and network analysis. The topic of the article is timely, the methods and software described/employed are sound, and the protocol presented is of interest for current research on transcriptomics. The manuscript is in general well written, and I believe it will be a valuable contribution to Methods and Protocols.

I only have a major concern about the focus of the study that is reflected in the title, and which I think must be modified. The protocol described skips/neglects basic steps of RNAseq analysis, which indeed begin with the processing of raw reads from a high-throughput sequencing experiments (including quality control, filtering, etc.), assembly of the transcripts (either de novo or referenced) and their validation/filtering. Only after that is accomplished, one can begin with the analysis described in this protocol paper or additional ones (functional annotation, variant calling, etc.).

As the author states himself (line 49), his protocol begins with data provided by a commercial sequencing service (Novogene in this case, but could be Macrogen, BGI... the list is endless) and then the protocol is mainly focused on different aspects of pathway analysis. However, just reading the title, one can have the impression that the protocol described can be useful also for the previous steps (raw reads processing, assembly) that are indeed left out. For people doing comparative/evolutionary research (gene discovery in non-model organisms [e.g. https://doi.org/10.1093/dnares/dsy034], phylotranscriptomics [e.g. https://doi.org/10.1038/s41559-017-0240-5], etc.), such previous steps are critically important as well, and the claim of the title of the paper becomes misleading because the protocol is not for general processing of RNAseq data (i.e. comprehensively including all steps from raw reads on), but has a main focus on the final-most steps (gene ontology, enrichment, pathway and network analyses). There are indeed other papers published with methods/protocols for comprehensive RNAseq data processing including all steps, such as TRUFA (https://doi.org/10.4137/EBO.S23873) or Galaxy (https://doi.org/10.1093/nar/gky379). Probably these should be cited in the manuscript as well.

Hence, I strongly believe that the title needs to be changed to reflect this situation. I would recommend a title like this (which more precisely describe what the protocol is for): “A Streamlined Approach to Pathway Analysis from RNA-sequencing Data”. Other than that, I think the paper is alright and merits publication. A few minor points follow:

- Line 24: RNA-sequencing is commonly abbreviated RNAseq (or RNA-seq). I think it would be appropriate to include this here in parenthesis.

- Line 67: something missing between “will” and “between” at the end of the line.

- Line 71-73: please explain why/how.

- Lines 82-85: please add reference(s) for this statement.

- Line 125: “be” missing after “can”.

- Line 129 (and throughout the text): rather than referring to Microsoft Excel, it would be more appropriate to refer to spreadsheets in general.

- Line 139: “be” missing after “can”.

- Line 152: change “discusses” to “discussed”.

- Lines 157-159: please explain why.

- Line 190: please provide the reference/URL for the BioGrid database.

- Line 229: something missing between “best” and “the” (display? show?).

- Line 304: insert “the” before “future”.

- Line 454: which “analysis tools” are you referring to? Please be more specific.

I hope the author finds my comments helpful.

Author Response

Bow – mps-1125050

Response to Reviewer 1:

Comments and Suggestions for Authors

This is a relatively straightforward manuscript that presents a processing protocol of RNASseq data for gene ontology, pathway and network analysis. The topic of the article is timely, the methods and software described/employed are sound, and the protocol presented is of interest for current research on transcriptomics. The manuscript is in general well written, and I believe it will be a valuable contribution to Methods and Protocols.

I only have a major concern about the focus of the study that is reflected in the title, and which I think must be modified. The protocol described skips/neglects basic steps of RNAseq analysis, which indeed begin with the processing of raw reads from a high-throughput sequencing experiments (including quality control, filtering, etc.), assembly of the transcripts (either de novo or referenced) and their validation/filtering. Only after that is accomplished, one can begin with the analysis described in this protocol paper or additional ones (functional annotation, variant calling, etc.).

As the author states himself (line 49), his protocol begins with data provided by a commercial sequencing service (Novogene in this case, but could be Macrogen, BGI... the list is endless) and then the protocol is mainly focused on different aspects of pathway analysis. However, just reading the title, one can have the impression that the protocol described can be useful also for the previous steps (raw reads processing, assembly) that are indeed left out. For people doing comparative/evolutionary research (gene discovery in non-model organisms [e.g. https://doi.org/10.1093/dnares/dsy034], phylotranscriptomics [e.g. https://doi.org/10.1038/s41559-017-0240-5], etc.), such previous steps are critically important as well, and the claim of the title of the paper becomes misleading because the protocol is not for general processing of RNAseq data (i.e. comprehensively including all steps from raw reads on), but has a main focus on the final-most steps (gene ontology, enrichment, pathway and network analyses). There are indeed other papers published with methods/protocols for comprehensive RNAseq data processing including all steps, such as TRUFA (https://doi.org/10.4137/EBO.S23873) or Galaxy (https://doi.org/10.1093/nar/gky379). Probably these should be cited in the manuscript as well.

Hence, I strongly believe that the title needs to be changed to reflect this situation. I would recommend a title like this (which more precisely describe what the protocol is for): “A Streamlined Approach to Pathway Analysis from RNA-sequencing Data”. Other than that, I think the paper is alright and merits publication. A few minor points follow:

Author Response:

Thank you very much for your comments on this manuscript, they have been very helpful.  I agrees with your concern that the title of this submitted work may be misleading in its scope, and therefore the title has been altered to the suggested title for clarity.  Additionally, the described references for TRUFA and Galaxy have been mentioned in the introduction to further reflect the intended scope of this methods manuscript.  Please find responses to your specific comments below, and I would like to again extend my appreciation for your time in reviewing this manuscript. 

Specific comments:

C1:  Line 24: RNA-sequencing is commonly abbreviated RNAseq (or RNA-seq). I think it would be appropriate to include this here in parenthesis.

R1:  The abbreviation “RNA-seq” has been added.

C2:  Line 67: something missing between “will” and “between” at the end of the line.

R2:  The described line has been corrected.

C3:  Line 71-73: please explain why/how.

R3:  The author agrees that further explanation would aid in clarity.  To address this, a reference discussing the risks associated “fishing expeditions” when processing omic data has been added, along with a general example of the rationale for narrowing the focus of analysis.

C4:  Lines 82-85: please add reference(s) for this statement.

R4:  To address this concern, references discussing fold-change threshold determination have been added.  Additionally, a statement has been added to clarify that the reason for a 2 fold-change threshold being selected in manuscript examples is based on Novogene recommendation.

C5:  Line 125: “be” missing after “can”.

R5:  The described line has been corrected.

C6:  Line 129 (and throughout the text): rather than referring to Microsoft Excel, it would be more appropriate to refer to spreadsheets in general.

R6:  All instances of “Excel” have been removed and replaced with the general term “spreadsheet(s)”.

C7:  Line 139: “be” missing after “can”.

R7:  The described line has been corrected.

C8:  Line 152: change “discusses” to “discussed”.

R8:  The described line has been corrected.

C9:  Lines 157-159: please explain why.

R9:  To address the described concern, a reference discussing the use of FDR values in RNA-seq has been included, as well as a statement to further clarify the importance of these values.

C10:  Line 190: please provide the reference/URL for the BioGrid database.

R10:  A reference for the use of the BioGrid database in RNA-seq analysis has been included.  Additionally, the URL for this database site exists within the procedural steps.

C11:  Line 229: something missing between “best” and “the” (display? show?).

R11:  The described line has been corrected.

C12:  Line 304: insert “the” before “future”.

R12:  The described line has been corrected.

C13:  Line 454: which “analysis tools” are you referring to? Please be more specific.

R13:  To address the described concern, the author has specified the analysis tools in this statement.

Reviewer 2 Report

In this manuscript, Bow provided a step-by-step workflow for researchers to handle RNA-sequencing data for downstream analysis, particularly to the gene set enrichment and pathway/network analysis. The manuscript provided a detailed introduction and analysis protocol, facilitating researchers to perform RNA-seq data analysis independently. However, the manuscript lacks clarity and other information for downstream analyses.

Specific comments:

  1. There are some assumed knowledges in the protocol where researchers with minimal bioinformatics knowledge will struggle to follow the data analysis independently. For example, different formats of RNAseq data can be generated. For example, .TXT file containing raw counts or .TXT file containing normalised counts or log counts. Other formats such as .CSV file. Normalised data should be used for downstream analysis. Could the author clarify?
  2. Additionally, the title of the manuscript is misleading as the downstream analyses of RNA-seq data include but not limited to differential gene expression analysis, immune deconvolution, and immunophenogram etc. At this point, the title of the protocol is not informative for other researchers. For example, researchers may expect the protocol to include the working steps from FASTQ files to pathway analysis. Could the author clarify?
  3. The author has covered a wide range of pathway analysis software. But there are some common user-friendly tools that were missed out such as FunRich () and GSEA MSigDB (cited > 2500 times). In addition, R package with shiny app was not included in the manuscript. Could the author clarify?
  4. There are also some important steps that were missing in the instructions. For example, DAVID requires the selection of species (although it is shown on the figure, it is not explained in text) and as well as the types (e.g. Gene Symbol, Ensembl ID etc.). Furthermore, DAVID included other pathway databases such as KEGG or Reactome etc. where users can select which database to use. However, this was not reflected in the protocol.
  5. In addition to pathway analysis, it is quite often that researchers also investigate biological processes, molecular function and cell components. However, this was not described in the manuscript. Without knowing the biological processes/molecular functions, researchers may have difficulty interpreting the pathways in their studies.
  6. The figures for the procedure steps are very helpful. A flow chart of the workflow for each tool will be useful and handy for researchers. The author could also include a decision tree workflow. This can be included in the supplementary material if needed.

Author Response

Bow – mps-1125050

Response to Reviewer 2:

Comments and Suggestions for Authors

In this manuscript, Bow provided a step-by-step workflow for researchers to handle RNA-sequencing data for downstream analysis, particularly to the gene set enrichment and pathway/network analysis. The manuscript provided a detailed introduction and analysis protocol, facilitating researchers to perform RNA-seq data analysis independently. However, the manuscript lacks clarity and other information for downstream analyses.

Author Response:

Thank you very much for your comments on this manuscript, they have been very helpful.  I agree with your concern that the title of this submitted work may be misleading in its scope, and therefore the title has been altered for clarity.  The revised title of the manuscript is based on a suggestion from the other reviewing party for this manuscript, and I believe that this new title is more reflective of the intended scope for this protocol.  As the target audience for this methods manuscript are individuals with limited experience with pathway analysis, I appreciate the guidance your comments have provided.  The primary objective of this article is to provide researchers that may be newer to bioinformatics with a template for pathway analysis of normalized RNA-seq data, which can then be modified and optimized for their particular experiment set-up.  Please find responses to your specific comments below, and I would like to again extend my appreciation for your time in reviewing this manuscript. 

Specific comments:

C1:  There are some assumed knowledges in the protocol where researchers with minimal bioinformatics knowledge will struggle to follow the data analysis independently. For example, different formats of RNAseq data can be generated. For example, .TXT file containing raw counts or .TXT file containing normalised counts or log counts. Other formats such as .CSV file. Normalised data should be used for downstream analysis. Could the author clarify?

R1:  The author agrees that the initial data being utilized in this protocol are normalized differentially expressed gene lists, as opposed to raw RNA-seq data.  To address this, text has been altered in the introduction (Lines 40-52) to better establish the initial data for the described method.

C2:  Additionally, the title of the manuscript is misleading as the downstream analyses of RNA-seq data include but not limited to differential gene expression analysis, immune deconvolution, and immunophenogram etc. At this point, the title of the protocol is not informative for other researchers. For example, researchers may expect the protocol to include the working steps from FASTQ files to pathway analysis. Could the author clarify?

R2:  As mentioned above, the title of the manuscript has been altered to better reflect the scope of the described protocol.  The new title, “A Streamlined Approach to Pathway Analysis from RNA-sequencing Data,” was suggested by the other reviewing party for this manuscript, as they shared this concern as well.

C3:  The author has covered a wide range of pathway analysis software. But there are some common user-friendly tools that were missed out such as FunRich () and GSEA MSigDB (cited > 2500 times). In addition, R package with shiny app was not included in the manuscript. Could the author clarify?

R3:  The author would like to thank the reviewer for listing these valuable analytical software options, and these will certainly be explored by the author.  The described protocol in this manuscript was derived from methodologies used for an on-going RNA-seq study and was compiled based on pathway analysis software with which the author was most familiar.  Additional user-friendly software tools, such as those mentioned in this comment, can certainly be utilized and would enhance the accuracy of downstream pathway selection.  To reflect this, a statement has been included to inform readers that other effective software tools exist outside of those described in this text, and that these tools, such as those listed in the above comment, can be further implemented or exchanged with the tools in this manuscript to optimize this workflow for researchers (Lines 137-141).

C4:  There are also some important steps that were missing in the instructions. For example, DAVID requires the selection of species (although it is shown on the figure, it is not explained in text) and as well as the types (e.g. Gene Symbol, Ensembl ID etc.). Furthermore, DAVID included other pathway databases such as KEGG or Reactome etc. where users can select which database to use. However, this was not reflected in the protocol.

R4:  The author agrees that further clarification of the capabilities of DAVID are needed.  The text has been altered to reflect this by noting that DAVID offers the ability to enrich data using both KEGG and Reactome databases (Lines 171-172) and by adding protocol lines within the section describing DAVID use (Lines 342-345).  For the DAVID analysis procedural steps, the described protocol focuses on the use of the default analysis parameters, though it is mentioned that these can be modified to for optimization.

C5:  In addition to pathway analysis, it is quite often that researchers also investigate biological processes, molecular function and cell components. However, this was not described in the manuscript. Without knowing the biological processes/molecular functions, researchers may have difficulty interpreting the pathways in their studies.

R5:  The described methods comprise a means for generating pathways of interest from normalized differentially expressed genes derived from RNA-sequencing data.  The formulation of focal parameter sets during data processing steps serves to both minimize risk of “fishing expeditions” and to frame the downstream analysis within a specific context.  For example, restricting the pathway analysis to pathways and GO relating to osteogenic and cell adhesion for an experiment involving a treatment intended to impact osteo-differentiation.  To this end the described protocol aims to provide researchers with a set of pathways and GO of interest that can then be further examined to elucidate mechanisms responsible for the observed changes in the transcriptional landscape.  This downstream interpretation of biological processes associated with the generated list of PoIs would in this case be outside of the intended scope for this manuscript.

C6:  The figures for the procedure steps are very helpful. A flow chart of the workflow for each tool will be useful and handy for researchers. The author could also include a decision tree workflow. This can be included in the supplementary material if needed.

R6: The author agrees that an overview flowchart and decision tree would be very helpful for readers and a great addition.  A new figure to address this has been added and designated as workflow overview (Figure 7).  Thank you for this suggestion.

Round 2

Reviewer 2 Report

Could the author cite the software/analysis tools mentioned in the manuscript? This also includes FunRich and GSEA mutsigdb in the revised manuscript.

(1) FunRich: An open access standalone functional enrichment and interaction network analysis tool
Proteomics. 2015 Aug;15(15):2597-601. doi: 10.1002/pmic.201400515. Epub 2015 Jun 17.

(2) Gene set enrichment analysis: A knowledge-based approach for interpreting genome-wide expression profiles
https://doi.org/10.1073/pnas.0506580102

Author Response

The author would like to thank the reviewer for their valuable comment.  To address the noted concerns additional references have been added to the manuscript for software/analysis tools.  Below are listed the references associated with each software/analysis tool:

  • SynGo: Reference #6
  • KEGG: Reference #7-8
  • WikiPathway: Reference #9-10
  • PID: Reference #11
  • Reactome: Reference #12
  • GO Consortium: Reference #13-14
  • FunRich: Reference #15
  • GSEA: Reference #16
  • IMPaLA: Reference #17
  • KOBAS: Reference #20
  • DAVID: Reference #24
  • CPDB: Reference #28-30
  • Cytoscape: Reference #37-38

Thank you again for your time in reviewing this manuscript and your valuable comments.

Best Regards,

Austin Bow
